# Biodegradation of the Antiretroviral Tenofovir Disoproxil by a Cyanobacteria/Bacterial Culture

**DOI:** 10.3390/toxics12100729

**Published:** 2024-10-10

**Authors:** Sandra Regina Silva, Gabriel Souza-Silva, Carolina Paula de Souza Moreira, Olívia Maria de Sousa Ribeiro Vasconcelos, Micheline Rosa Silveira, Francisco Antonio Rodrigues Barbosa, Sergia Maria Starling Magalhães, Marcos Paulo Gomes Mol

**Affiliations:** 1Faculdade de Farmácia, Universidade Federal de Minas Gerais, Belo Horizonte 30510010, Brazil; srssandrars@gmail.com (S.R.S.); silva_gs@yahoo.com (G.S.-S.); michelinerosa@gmail.com (M.R.S.); barbosa.ufmg@gmail.com (F.A.R.B.); sergia.starling@gmail.com (S.M.S.M.); 2Fundação Ezequiel Dias, Departamento de Pesquisa e Desenvolvimento, Belo Horizonte 30510010, Brazil; carolina.moreira@funed.mg.gov.br (C.P.d.S.M.); oliviamrv@gmail.com (O.M.d.S.R.V.)

**Keywords:** drug, *Microcystis novacekii*, *Pseudomonas pseudoalcaligenes*, by-products, tenofovir isoproxil monoester

## Abstract

Tenofovir disoproxil fumarate (TDF) is an antiretroviral drug extensively used by people living with HIV. The TDF molecule is hydrolysed in vivo and liberates tenofovir, the active part of the molecule. Tenofovir is a very stable drug and the discharge of its residues into the environment can potentially lead to risk for aquatic species. This study evaluated the TDF biodegradation and removal by cultures of *Microcystis novacekii* with the bacteria *Pseudomonas pseudoalcaligenes*. Concentrations of TDF of 12.5, 25.0, and 50.0 mg/L were used in this study. The process occurred in two stages. In the first 72 h, TDF was de-esterified, forming the tenofovir monoester intermediate by abiotic and enzymatic processes associated in an extracellular medium. In a second step, the monoester was removed from the culture medium by intracellular processes. The tenofovir or other by-products of TDF were not observed in the test conditions. At the end of the experiment, 88.7 to 94.1% of TDF and its monoester derivative were removed from the culture medium over 16 days. This process showed higher efficiency of TDF removal at the concentration of 25 mg/L. Tenofovir isoproxil monoester has partial antiviral activity and has shown to be persistent, maintaining a residual concentration after 16 days in the culture medium, therefore indicating the need to continue research on methods for total removal of this product from the aquatic environment.

## 1. Introduction

In recent decades, problems related to water contamination have become a global concern. In this context, the dispersion of pharmaceutical residues in the aquatic environment has been the target of numerous studies [1,2,3]. Drugs are biologically active molecules and their unpredictable effects on species exposed in the aquatic environment may represent a potential risk to aquatic ecosystems [4,5].

Studies have indicated the presence of drug residues in several environmental matrices, such as soil, sewage, surface, and treated waters [1,2,3,4,5,6]. Among the drugs, antiretrovirals (ARVs) are of particular concern, since some of them act in viral DNA synthesis and, potentially, may have effects on the replication process of other organisms.

According to United Nations data on human immunodeficiency virus (HIV) [7], currently 37.7 million people worldwide are living with HIV (PLHIV), and the control of the pandemic depends on continuous antiretroviral therapy. Thus, due to its widespread use, the presence of ARVs in the environment have been reported in different countries [8,9,10,11].

Tenofovir disoproxil fumarate (TDF) is an ARV of great importance because it is one of the first-line drugs, combined with other antiretroviral drugs, in HIV/AIDS treatment. Tenofovir is an analogue of the nucleotide adenosine 5′-monophosphate and acts by inhibiting the reverse transcriptase enzyme necessary for viral replication in human cells. Due to the low lipid solubility [12] of tenofovir, it is administered as a prodrug, in esterified form, and it is de-esterified intracellularly to release the active portion of the molecule, tenofovir [13,14] (Figure 1).

About 80% of the dose of TDF administered to humans is eliminated in the active form, as a de-esterified product, tenofovir (Figure 1), without undergoing further metabolism [12]. The tenofovir molecule is very stable [15], which makes the treatment for its removal from the aquatic environment a challenge, both in the case of industrial effluents and domestic sewage.

Effluent treatment technologies that employ microorganisms are the most used processes for the degradation of organic compounds [16]. In recent years, the use of cyanobacteria and microalgae in waste treatment has aroused interest [16,17,18]. These are photosynthetic organisms and the presence of these groups in consortia makes the process more sustainable compared to technologies that use bacteria intercropping.

Cyanobacteria species also have the ability to degrade different types of chemical compounds by mixotrophic metabolisms acting together with bacteria in the degradation of chemicals in the environment [19,20].

Cyanobacteria of the genus Microcystis have presented interesting results in biodegradation studies of several substrates. These species show resistance and rapid responses to eutrophication [21], possibly by acclimation or adaptation to the adverse conditions [22].

Microcystis is one of the most commonly found genera in Brazil [23], which justifies evaluating the potential of Microcystis species locally for biodegradation processes. Cultures of *Microcystis novacekii* have been studied and have demonstrated their ability to remove different types of pollutants [24,25].

Cyanobacteria usually present themselves in the environment associated with bacteria through interactions considered mutualistic, where cyanobacteria provide heterotrophic bacteria with organic matter to sustain their growth and oxygen for greater efficiency of aerobic metabolism. At the same time, the bacteria provide the cyanobacteria with carbon dioxide and essential nutrients for their metabolism. Thus, these associations are fundamental for the cycling of carbon and other essential ions in the aquatic environment [26,27,28]. The mutualistic interactions between these organisms make it very difficult to maintain axenic cyanobacterial strains in biodegradation studies [27,29].

There are no studies in the literature on the removal of TDF from the aquatic environment by cyanobacteria culture, justifying the study of the potential of a *M. novacekii* culture to remove this antiviral. Therefore, the objective of this study was to evaluate the process of biodegradation of tenofovir disoproxil using a non-axenic culture of *M. novacekii* isolated from a Brazilian lake.

## 2. Materials and Methods

### 2.1. Reagents and Chemicals

The tenofovir disoproxil fumarate used in the experiments was obtained from Nortec Química (Duque de Caxias, Rio de Janeiro, Brazil) (lot 507034) as a white, amorphous solid. The drug was analysed and certified by the Department of Quality Control of the Ezequiel Dias Foundation (FUNED). The reagents, solvents, and other chemicals used were of analytical or high-performance liquid chromatography (HPLC) grade. All the solutions were prepared using purified water.

### 2.2. Microcystis novacekii Culture

A cyanobacterium strain, *M. novacekii* (Komárek) Compère, was isolated from water samples collected in Dom Helvécio Lake, in the Rio Doce State Park (42°35′595′′ S; 19°46′419′′ W), Minas Gerais, Southeastern Brazil) in May 2004. For the isolation of *M. novacekii* strains, pipetting and dilution series of lake water samples were used. The species was isolated in its pure form and cultivated in WC (Wright’s cryptophytes) culture medium [30,31].

The procedure rendered a non-axenic unialgal culture which was kept in the culture with a WC medium at pH 7 and a controlled temperature of 25 ± 2 °C under a fluorescent light of 110.5 µmol photons m^2^/s and a 12 h photoperiod. The non-axenic *M. novacekii* strain was kept in culture in the algae and cyanobacteria bank of the Laboratory of Limnology, Ecotoxicology and Aquatic Ecology at the Institute of Biological Sciences of the Federal University of Minas Gerais (LIMNEA-ICB-UFMG).

The *Microcystis novacekii* strain was tested for the presence of microcystin to verify the toxigenic potential, using enzyme-linked immunosorbent assays and the polymerase chain reaction to amplify the mcyB gene in the DNA of this strain. Both results were negative for microcystin. During the experiments, the presence of *Pseudomonas pseudoalcaligenes* was observed in the culture. The bacterium associated with *M. novacekii* was identified by Neoprospecta Microbiome Technologies Company using next-generation amplicon sequencing (NGS) and Neobiome software.

### 2.3. Microcystis novacekii Culture Medium

ASM-1 medium [32] buffered by the addition of 750 mg/L of 3-(N-morpholino) propanesulfonic acid (MOPS), pKa 7.2, was used for *M. novacekii* culture. The pH was adjusted to 7.0 with 0.1 mol/L HCl or NaOH solution. The composition of the ASM-1 medium was (mg/L): NaNO_3_ (170.00), CaCl_2_ (29.00), MgCl_2.6_H_2_O (41.00), MgSO_4_ (49.00), K_2_HPO_4_ (8.70), Na_2_HPO_4.12_H_2_O (17.80), CoCl_2.6_H_2_O (9.5 × 10^−3^), CuCl_2.2_H_2_O (6.5 × 10^−4^), MnCl_2.4_H_2_O (0.69), ZnSO_4.7_H2O (0.35), H_3_BO_3_ (1.24), FeCl_3.3_H_2_O (0.54), and EDTA Na_2_ (3.72).

### 2.4. TDF Biodegradation by M. novacekii

The pre-culture of *M. novacekii* was prepared by inoculation of *M. novacekii* to a flask containing ASM-1 medium. This flask was incubated under controlled temperature (25 ± 2 °C). Cell density was evaluated by optical density (680 nm) [33] and was monitored until the culture reached approximately 10^6^ cell/mL. The TDF biodegradation tests in culture of the cyanobacterium *M. novacekii* were carried out following the OECD standardized protocol (ready biodegradability tests) n° 301—Guidelines for testing of chemicals (2003) with adaptations [34]. The TDF solution (2.5 g/mL) was added to test flasks containing 100 mL of *M. novacekii* culture with a cell density of approximately 106 cell/mL to obtain final concentrations of 50.00, 25.00, and 12.50 mg/L. Cyanobacteria culture was used as growth control. To determine the stability of TDF in ASM-1 medium, an uninoculated sterile medium in the same concentrations of the test flasks was prepared. The test flasks and controls were transferred to a shaking table and were incubated for 16 days, under controlled temperature (25 ± 2 °C) and 5000 lx (c. 11 W/m^2^) illumination from cool-white fluorescent lamps. All experiments were conducted in triplicate.

### 2.5. Extraction of TDF and Its Metabolites from Culture Medium

After 1, 3, 7, and 16 days, a 5.00 mL sample was removed from each test flask, filtered (0.45 µm—Millipore), and the aqueous portion was submitted to solid-phase extraction. The samples were subjected to solid-phase extraction using a Phenomenex Strata-X^®^ cartridge (Phenomenex, Torrance, CA, USA). The cartridges were conditioned with 5 mL of purified water. The samples were transferred to the cartridge and eluted with 5 mL of 5% methanol solution, followed by 5 mL pure methanol. The eluate was filtered (Millipore Filter, 0.22 μm pore size), transferred to a vial with an insert, and injected into a liquid chromatography equipment coupled to a mass spectrophotometer for HPLC quadrupole time-of-flight mass spectrometry (HPLC/Q-TOF-MS) analysis.

### 2.6. HPLC/Q-TOF-MS HPLC Analysis

A high-pressure liquid chromatography coupled with quadrupole time-of-flight mass spectrometry (HPLC–QTOF-MS) system (6540 UHD Accurate Mass Q-TOF LC/MS equipped with Agilent Mass Hunter Workstation Data Acquisition software B.06.00) was used and a Zorbax Eclipse Plus C18 column (2.1 × 50 mm; particle size 1.8 mm), with the following experimental conditions: flow rate of 0.5 mL/min; mobile phase methanol:water, both with 0.1% formic acid in gradient elution (50% of methanol for 2 min and 50–100% methanol for 3 min, and then returned to 50% of methanol for 1.5 min, totalling 6.5 min); injection volume of 4 µL, temperature of 50 °C. ESI parameters were capillary voltage 3.5 kV for positive mode; gas temperature 325 °C; drying gas 8 L/min; fragmentor 175 V; skimmer 65 V; mass range from 100 to 1000 *m*/*z* and no collision energy was used. The TDF calibration curve was prepared in triplicate in water, with concentrations ranging from 0.1 to 125 mg/L, followed by filtration (0.22 μm).

## 3. Results and Discussion

The analytical method for monitoring the experiments was developed using an aqueous solution of TDF (50.0 mg/L), analysed via HPLC-ESI-Q-TOF/MS. The presence of two peaks (Figure 2), a main peak (0.7 min) and a residual peak (0.3 min) was observed in the chromatogram. For peak characterization, the isolated ions in the mass spectrum were extracted and the corresponding chemical structure was proposed based on the exact mass obtained. Thus, the peak at 0.3 min, with *m*/*z* 404.13, according to the mass fragmentation pattern described by Kurmi et al. [35], was assigned to tenofovir isoproxil protonated monoester (TMF). The peak at 0.7 min, with *m*/*z* 520.18 was attributed to protonated tenofovir disoproxil TDF [M + H +]. In the final elution time (after 4 min), no TDF-derived fragment was observed, with the final peaks corresponding to the gradient elution.

According to Kurmi et al. [35], the tenofovir isoproxil monoester detected in the medium corresponds to the product of partial hydrolysis of TDF and is described as an impurity commonly present in the raw material of the drug [36].

To carry out the tests, the contribution of abiotic processes to the degradation of TDF in ASM-1 medium was evaluated, under controlled conditions (pH, temperature, and light radiation). It was observed in the controls (TDF concentrations 12.5, 25, and 50 mg/L) an increase in the peak attributed to the monoester throughout the experiment. At the end of the process (16 days), all the TDF was converted into tenofovir isoproxil monoester, showing that mono-de-esterification can occur spontaneously in the culture medium.

Hydrolysis was observed at all TDF concentrations used. ASM-1 medium is a mineral medium containing several salts, and the experiment was performed in buffered medium (pH 7–8). According to Silva [37], the hydrolysis of TDF occurs preferably at a neutral or alkaline pH, with the molecule being more stable at an acidic pH (2 to 3). Thus, the test conditions favored the de-esterification of the molecule.

Biodegradation tests were performed using TDF concentrations of 12.5, 25, and 50 mg/L. It is important to highlight that this series of concentrations were defined considering that the ability of strains of microorganisms to degrade toxic compounds depends on their intrinsic ability to metabolise the xenobiotic. This property can be constitutive or acquired by microorganisms exposed to conditions considered stressful for the species [22,38].

Microbial biochemical pathways can be activated when microorganisms are exposed to critical conditions, such as high concentrations of pollutants, temperature, and pH variations [22,39]. In the case of *M. novacekii*, in a previous study [40], the strain tolerated high concentrations of TDF (EC50% 161.0 mg/L), allowing the use of drug concentrations of up to 50 mg/L in this study, configuring a stressful condition to evaluate the metabolization potential of this antiviral by the culture of *M. novacekii*.

During the TDF biodegradation experiments using the *M. novacekii* culture, the drug’s mono-de-esterification was also verified; however, the hydrolysis in the tests was more intense and faster compared to controls (Figure 3). While in the control samples total mono-de-esterification occurred over 16 days, in cultures, after 72 h, only TDF residues were detected, indicating that although abiotic factors contribute to hydrolysis, metabolic pathways of microorganisms are probably responsible for accelerating the de-esterification process through extracellular enzymes.

It was verified during the test that the drug and its derivative were gradually extracted from the medium, with a percentage of TDF/TMF removal at the end of the experiment of 91.8% (12.5 mg/L), 94.1% (25 mg/L), and 88.7% (50 mg/L). In controls, no reduction in monoester concentration was observed (Figure 4). Neither tenofovir, nor any other metabolite besides TMF was detected in the culture medium, indicating that the removal of TMF is probably due to the direct action of microorganisms.

To analyze these results, it is necessary to consider that the culture of *M. novacekii* employed is unialgal, but not axenic. According to the genetic sequencing performed, the bacterium *P. pseudoalcaligenes* was identified in the medium. This is an aerobic, Gram-negative species and its potential to metabolise toxic compounds has been described by several researchers [41,42,43]. Its production of esterases stands out, including potent arylesterases that give this species the ability to degrade various compounds, including polyesters [41,42]. The production of these esterases may be the accelerating factor of TDF hydrolysis.

The coexistence of cyanobacteria and microalgae with heterotrophic bacteria in the environment has aroused interest in the study of degradation of organic substances, leading to the inclusion of photosynthetic species in microbial consortia. The association of these groups of organisms can be advantageous for reducing energy expenditure due to in situ oxygen production, while also reducing CO_2_ emissions and increasing the production of algal biomass that can be used in the production of various compounds of technological application [16,44]. Thus, several studies of the potential of these associations for the removal of different classes of organic substances have been carried out [45,46,47].

Phytoplanktonic species show similar behaviour in the face of stressors [48], which may increase the expression of enzymes degrading organic compounds and other compounds aimed at cellular protection [18].

In the study of microbial associations for the biodegradation of pollutants, the presence of species of the genus Microcystis is particularly interesting as they are very resistant to toxic agents due to protection mechanisms developed throughout the evolutionary process [38]. In the case of the genus Microcystis, resistance has been attributed to the characteristics of some species, including those that present a thick mucilaginous layer that surrounds the cells, with diverse functions such as nutrition and protection against dissection and against external agents, in addition to allowing the aggregation of cells in colonies that favor the formation of biofilm [17,38].

The role of cell protection by mucilage is highlighted by Pugnetti et al. [49] who reported that when exposed to adverse conditions such as the presence of toxic substances, some species intensify the production of the mucilaginous layer, which behaves like a sponge that absorbs xenobiotics [50]. In general, mucilage can biosorb xenobiotics through different interactions, usually weak bonds, without energy consumption and can retain metal ions, natural organic matter, and toxic organic substances [16,50]. Chan et al. [16] reported that microalgae can biosorb compounds rich in nitrogen, phosphorus, heavy metals, antibiotics, organochlorines, pesticides, and azo dyes from aqueous matrices.

The presence of mucilage is one of the factors that facilitates the association with bacteria, since the exopolymers that compose the mucilage can be used for fixation and as a nutritional source to bacteria, offering heterotrophic microorganisms an ideal microenvironment for their growth and metabolism [51]. Thus, in the associations of cyanobacteria and heterotrophic bacteria, two processes can act in the removal of pollutants, the immobilisation of toxic substances by mucilage, and microbial degradation by heterotrophic bacteria and by cyanobacteria themselves [47].

In the proposition of cyanobacteria/microalgae and bacteria consortia, although the immobilisation by adsorption to mucilage allows for the reduction of soluble organic carbon, it is preferable to associate the mechanisms of biodegradation and especially of mineralization of pollutants as they are more effective for the removal of organic compounds. In general, biodegradation occurs through metabolic pathways typical of various microorganisms (bacteria, fungi, and algae), which can be expressed under stressful conditions. Biodegradation can lead to partial decomposition of the molecule, generating by-products or total decomposition, reducing organic compounds through metabolic processes to their inorganic forms [16].

Microalgae are organisms capable of degrading various types of substances into carbon sources and removing them from water. Several studies indicate the dehydration capacity of different species of microalgae, such as *Chlorella* spp., *Scenedesmus* spp., and *Aphanocapsa* spp. [52,53] through processes of biosorption and/or bioconversion and/or biodegradation. In addition, phytomediation with microalgae species is an effective way to ensure environmental sustainability, as they are widely distributed around the world, are good at degrading substances, have fast metabolism, and are cost-effective [18]. Cyanobacteria are prokaryotic microorganisms that thrive in a variety of conditions and have the potential to degrade substances while in search of sources to dispense and then use in their growth, stimulating water quality [54].

Studies of microalgae/cyanobacterial and bacterial consortia for the purification of drugs from the aquatic environment have been reported with promising results [17,20]. The use of these associations for the biodegradation of antimicrobials is highlighted. This is an important characteristic of cyanobacteria and microalgae given the difficulty in obtaining bacterial species tolerant to the biocidal effects of these compounds. Wang et al. [47], using a consortium of microalgae-heterotrophic bacteria for the degradation of chlortetracycline, observed that this drug was initially removed by biosorption, followed by biodegradation.

At the end of the experiment, the authors concluded that biosorption alone had a negligible contribution to the drug removal process, which does not mean that biosorption does not have an initial action on drug retention, facilitating the action of enzymes retained in the mucilaginous layer. Biodegradation catalysed by enzymes secreted by the species present, under stress triggered by the antibiotic, has been identified as the main mechanism of chlortetracycline removal [47]. These authors observed that the use of microalgae/heterotrophic bacteria cultures for chlortetracycline biodegradation presented better results in terms of drug bioremoval than the respective axenic cultures.

Likewise, in a study of the removal of sulfamethoxazole from the aquatic environment by a consortium of algae and heterotrophic bacteria, Rodrigues et al. [46] found that the antibiotic was mainly removed by biodegradation. The bioaccumulation and biosorption of the drug by the microorganisms were negligible. The small contribution of drug biosorption on microalgae cell walls was attributed to the high water solubility of the sulfamethoxazole molecule [46].

In this study, using the association of *M. novacekii* and *P. pseudoalcaligenes*, it is likely that the removal of TDF occurs in a similar way to that reported in the degradation of chlortetracycline [47] and sulfamethoxazole [46]. Possibly in the initial step of the process, mono-de-esterification occurs by an abiotic process and by the action of extracellular hydrolases from both heterotrophic bacteria and cyanobacteria, since both groups of microorganisms can be esterase producers [27,55].

The rapid formation of TMF and its slow removal from the medium, without the presence of other metabolites, suggests that degradation occurs in the intracellular medium and that penetration into cells, whether bacterial or cyanobacterial, is the critical factor for the slow removal of TMF from the medium. Khan et al. [56] described that during the proliferation stage of bacterial and microalgae associations, both groups express enzymes such as phosphatase, sulfatase, glucosidase, and galactosidase that may be responsible for biodegradation processes of organic compounds. Thus, it is not possible to state which of the groups of microorganisms was responsible for the removal of TMF.

The role of *M. novacekii* in the degradation of TMF is reinforced by the presence of phosphate groups in the molecule, a limiting nutrient for the growth of cyanobacteria. Ren et al. [57] found that *Microcystis aeruginosa* can use dissolved organic phosphate from different chemical compounds to support its growth. In this way, the TMF molecule can provide a source of phosphate for the cyanobacteria.

Although the removal of TDF from the medium occurred with high yield, the presence of residual concentrations of TMF at the end of the experiment was observed for all concentrations tested. At the concentration of 25 mg/L, the highest removal was obtained, about 94% of the drug and its metabolite, which is a very promising result. Apparently, intra- and extracellular conjugated processes occur, requiring further studies to optimize the drug extraction process and its metabolite from the medium.

Although TDF has a high efficiency for biodegradation, there is the possibility of improving strains of *M. novacekii* through genetic engineering, increasing the number of active groups in the cell wall to stimulate the attraction of pollutants, increasing the rate of biodegradation [58,59,60]. However, there are some barriers to the application of genetic engineering, such as regulatory restrictions and ecological concerns [18].

The method for biodegradability assessment used in this study is based on the di-rect measurement of the disappearance of the original organic compound through spe-cific chromatographic analysis to obtain information about primary biodegradation. This method is advantageous when compared to methods that use indirect measurements of bioconversion, such as CO₂, DOC, and BOD, as the use of only one indirect biodegradation parameter can provide misleading results [61].

Although biodegradation tests have been criticized, they are an important part of the information needed to predict the environmental fate of chemical products [62]. According to the OECD, TDF is classified as a readily biodegradable substance, as it achieved a sufficient extent of biodegradation in the OECD 301 test, respecting the 28-day time limit [34].

Thus, based on the OECD 301 series ready biodegradability tests, TDF is considered a “non-persistent” substance, as a positive result in a ready biodegradability test is suf-ficient to confirm non-persistence [62]. Additionally, taking into account the important observations made by Strotmann et al. (2023) [62], several important criteria were met to improve the quality of the biodegradation test, such as the assessment of the toxic effect of TDF on the cyanobacteria *M. novacekii* through the growth inhibition test [40].

Based on this criterion, it was possible to verify that there are no suspicions that TDF may have inhibitory effects on the inoculum in biodegradability tests, making it unnecessary to add inhibitory control vessels to avoid false negative conclusions [34,62,63].

Excluding the possible toxic effects of TDF on cyanobacteria [40], the reduction in biodegradation capacity with the increase in the concentration of the test substance may be a consequence of metabolic products formed during the biodegradation process and/or a change in the pH of the medium. Although traditional media used in OECD tests typically have a low phosphate concentration (3.7 mM), the culture medium used in this study was relatively high, approximately 18 mM. This increase in phosphate concentration results in good buffering properties, as demonstrated in a combined test system by Strotmann et al. (1995) [64], where the phosphate content was raised to 25.1 mM to allow stable buffering properties in the pH 7.0 range, which is necessary for biodegradation processes.

Thus, the production of metabolites, such as tenofovir and alkylphosphonate acids, during the biodegradation process can compromise the efficiency of the process [40]. However, many bacteria, such as *M. novacekii*, are capable of degrading not only the substances, but also their metabolites, forming smaller by-products, such as alcohols, oxygenated compounds, carbon, and water. During this biodegradation process, the breakdown of FTD causes a release of acids (such as carboxylic and phosphonic acids), which reduces the pH value.

However, even with the formation of possibly toxic by-products and the small change in pH, and even with the good results obtained in this study, it should be noted that tenofovir is an inhibitor of DNA synthesis, and its metabolite TMF is partially active [65], therefore its persistence in the environment can potentially lead to damage to the genetic heritage of exposed species. Thus, the persistence of TMF in culture medium for more than 15 days is worrying and points to the need for further studies on the biodegradation of this antiviral to prevent possible genotoxic actions to other aquatic organisms.

## 4. Conclusions

Through this study, excellent results were obtained in the removal of TDF from the culture medium using a culture of *M. novacekii* and *P. pseudoalkaligenes*. Approximately 94% of the drug and its metabolite were extracted at a concentration of 25 mg/L. After 16 days, residual concentrations of only one TDF metabolite, tenofovir isoproxil monoester, was detected. This result reinforces the potential of this association for studies on the removal of this drug from more complex matrices. The sustainability of the method, the ease of the technique and the good performance of the culture in removing TDF are advantages that justify the investigation of the potential of this association for environmental uses. The rapid de-esterification of TDF in the culture medium releasing the monoester—tenofovir isoproxil was one of the most important results of this study. TMF partially maintains the antiviral activity and the persistence of residual concentrations of this compound for more than 16 days in the culture medium is a worrying factor, as it may indicate that this metabolite can accumulate in the environment. In this sense, further studies aimed at removing this metabolite are necessary in order to prevent exposure of aquatic species to antiviral residues.

## Figures and Tables

**Figure 1 toxics-12-00729-f001:**
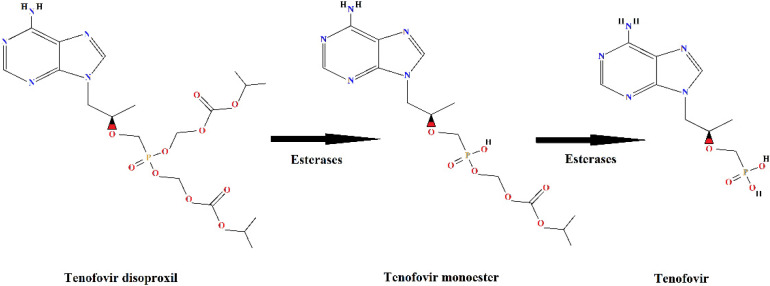
Metabolism of the prodrug tenofovir disoproxil with formation of the de-esterification products: tenofovir monoester and tenofovir.

**Figure 2 toxics-12-00729-f002:**
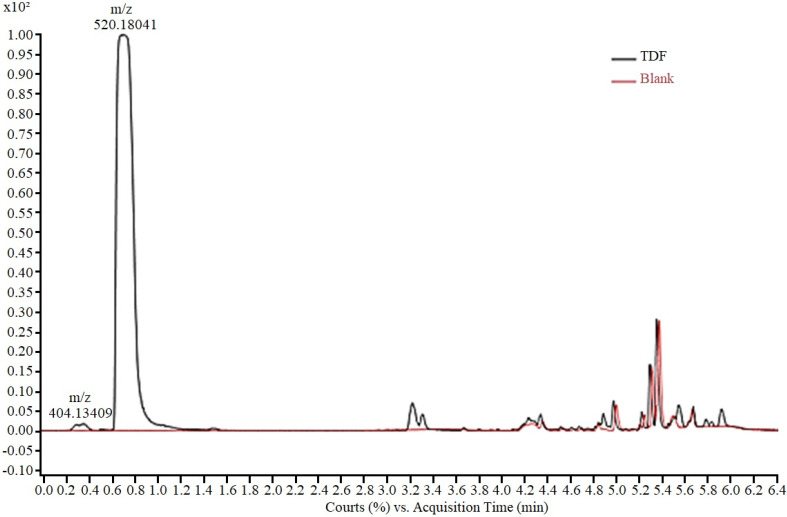
Chromatogram of TDF solution in ASM-1 medium (50 mg/L) showing peaks corre-sponding to tenofovir isoproxil monoester (*m*/*z* 404.13) and tenofovir disoproxil (*m*/*z* 520.18).

**Figure 3 toxics-12-00729-f003:**
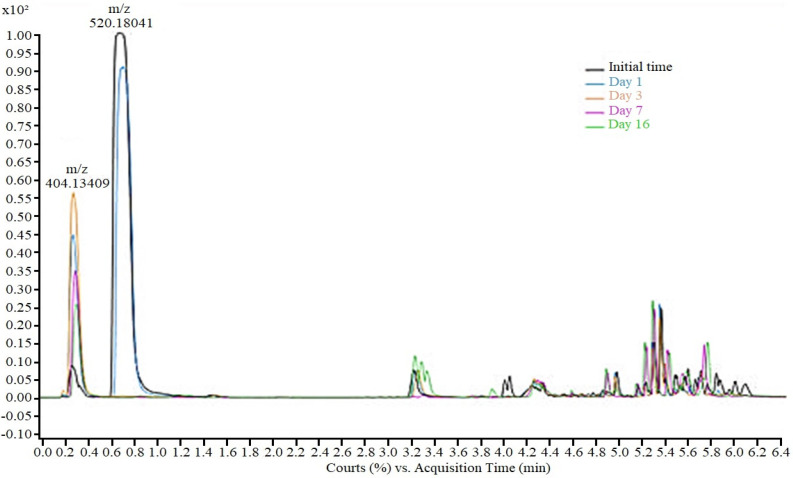
Tenofovir disoproxil and monoester of tenofovir removal during cyanobacteria degra-dation process in sample of 50 mg/L.

**Figure 4 toxics-12-00729-f004:**
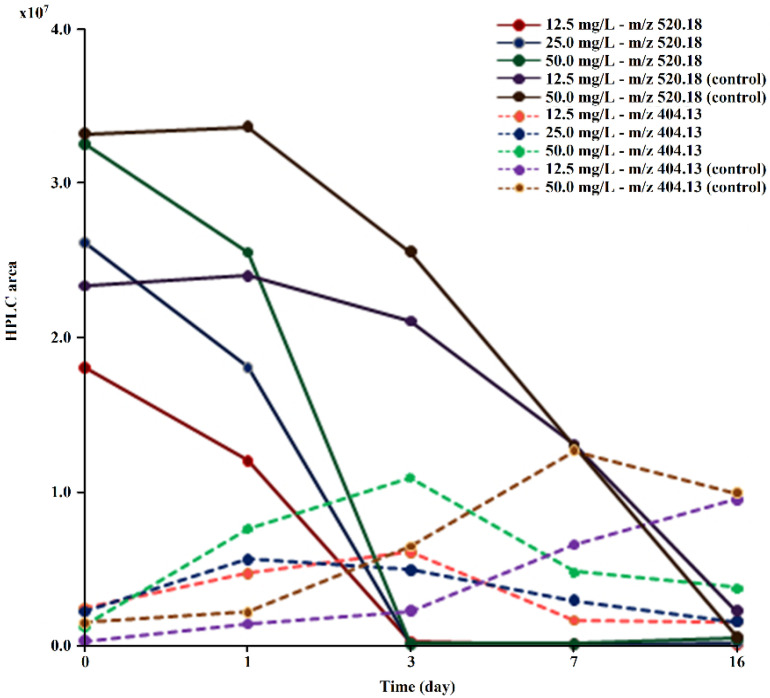
Evolution of the peak areas at *m*/*z* 520.18 and *m*/*z* 404.13 obtained via HPLC/MS during TDF (at concentrations of 12.5, 25.0, and 50.0 mg/L) biodegradation by *M. novacekii*.

## Data Availability

Data are contained within the article.

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
