# Peer review of "Biodegradation of the Antiretroviral Tenofovir Disoproxil by a Cyanobacteria/Bacterial Culture"

_toxics, 2024, doi:10.3390/toxics12100729_

Round 1

Reviewer 1 Report

Comments and Suggestions for Authors

In the manuscript "Biodegradation of the antiretroviral tenofovir disoproxil by a cyanobacteria/bacteria culture", the author evaluated the TDF biodegradation and removal by cultures of Microcystis novacekii with the bacteria Pseudomonas pseudoalcaligenes associated. This manuscript has guiding significance for the preparation of efficient TFNM for drug control in the water environment, but there are still some problems that need to be improved, so I recommend accepting it after major revision.

1.    All experimental data should contain error bars to reflect the repeatability of the experiment and the reliability of the data. Error bars can help readers better understand the variability of data and the statistical significance of experimental results.

2.    There is a lack of in-depth mechanism exploration of the degradation process, such as the analysis of degradation products and the prediction of degradation pathways.

3.    The manuscript simply lists the experimental results of the study, and the discussion of the mechanism is not enough.

4.    The literature cited in the manuscript is not rich enough and the logical connection between the literature is poor.

5.    The following literature is recommended for reference: Applied Catalysis B Environmental 311(7739):121363.

Comments on the Quality of English Language

Moderate editing of English language required.

Author Response

Date: 30th Sep/2024

Dear editors,

Thank you very much for reviewing our manuscript ID toxics-3227819, entitled "Biodegradation of the antiretroviral tenofovir disoproxil by a cyanobacteria/bacteria culture", submitted to Toxics journal. Your suggestions were valuable in contributing to our discussion of this topic.

The present letter provides point-by-point responses of the reviewer’ previous comments. We hope to have responded to all of them.

Kind regards,

Authors

Comments and Suggestions for Authors   

Reviewer 1:

In the manuscript "Biodegradation of the antiretroviral tenofovir disoproxil by a cyanobacteria/bacteria culture", the author evaluated the TDF biodegradation and removal by cultures of Microcystis novacekii with the bacteria Pseudomonas pseudoalcaligenes associated. This manuscript has guiding significance for the preparation of efficient TFNM for drug control in the water environment, but there are still some problems that need to be improved, so I recommend accepting it after major revision.

  1. All experimental data should contain error bars to reflect the repeatability of the experiment and the reliability of the data. Error bars can help readers better understand the variability of data and the statistical significance of experimental results.

A: Thank you very much for your valuable comments. We change all topics in the paper, as suggested. The results reflect the chromatography peaks generated during the tests, which indicate reductions in the concentrations of the active TDF evaluated. We opted for a cleaner graph to enhance visibility and make the data easier to interpret. Error bars were omitted to maintain this clarity.

  1. There is a lack of in-depth mechanism exploration of the degradation process, such as the analysis of degradation products and the prediction of degradation pathways.

A: We have worked to improve the discussion section by expanding on the details of the degradation process mechanisms. Additionally, we have suggested possible pathways for the degradation products and provided predictions for the degradation routes.

  1. The manuscript simply lists the experimental results of the study, and the discussion of the mechanism is not enough.

A: We have worked to improve the discussion section by expanding on the details of the degradation process mechanisms. Additionally, we have suggested possible pathways for the degradation products and provided predictions for the degradation routes.

  1. The literature cited in the manuscript is not rich enough and the logical connection between the literature is poor.

A: We included more scientific works aiming to improve the paper.

  1. The following literature is recommended for reference: Applied Catalysis B Environmental 311(7739):121363.

A: Thank you for this suggestion. We included more scientific works aiming to improve the paper.

Comments on the Quality of English Language

Moderate editing of English language required.

A: A grammatical review was conducted throughout the text with the collaboration of professional Marília Ribeiro de Vasconcelos.

Reviewer 2 Report

Comments and Suggestions for Authors

See comments

Comments on the Quality of English Language

The English language is understandable, but there are several flaws.

Author Response

Date: 30th Sep/2024

Dear editors,

Thank you very much for reviewing our manuscript ID toxics-3227819, entitled "Biodegradation of the antiretroviral tenofovir disoproxil by a cyanobacteria/bacteria culture", submitted to Toxics journal. Your suggestions were valuable in contributing to our discussion of this topic.

The present letter provides point-by-point responses of the reviewer’ previous comments. We hope to have responded to all of them.

Kind regards,

Authors

Comments and Suggestions for Authors   

Reviewer 2:

First of all, I want to state that the topic is very interesting and deserves publication. The methodology proposed by the authors using OECD based test systems is fine, but in the very end the authors should mention which test method was really used. Was it an OECD 301 test system for ready biodegradability or a test method for inherent biodegradability. There is a very detailed recent review covering this field (see: Strotmann, U., Thouand, G., Pagga, U. et al. Toward the future of OECD/ISO biodegradability testing-new approaches and developments. Appl Microbiol Biotechnol 107, 2073–2095 (2023). https://doi.org/10.1007/s00253-023-12406-6. In another recent review (see: Strotmann, U., Durand, MJ., Thouand, G. et al. Microbiological toxicity tests using standardized ISO/OECD methods—current state and outlook. Appl Microbiol Biotechnol 108,454 (2024). https://doi.org/10.1007/s00253-024-13286-0) the interplay of microbial toxicity and biodegradation is covered in detail. Also this review should be taken into account.

A: Thank you very much for your important comments. We change all topics in the paper, as suggested. We have worked to improve the discussion section by expanding on the details of the degradation process mechanisms. Additionally, we have suggested possible pathways for the degradation products and provided predictions for the degradation routes. We included more scientific works aiming to improve the paper.

These OECD test systems are based on DOC removal, oxygen consumption and carbon dioxide production to estimate ultimate biodegradability. In contrast, the investigations presented in this paper are very tentative and not based on well known OECD methods. There should be a clarification. I feel that the paper should be thoroughly overworked taking into account the points mentioned above. Therefore, a major revision is recommended.

A: We clarified the methods adopted and included information about methods, results and discussion in specifics paragraphs.

Round 2

Reviewer 1 Report

Comments and Suggestions for Authors

Accept in present form.

Reviewer 2 Report

Comments and Suggestions for Authors

I feel the article can be published in its present form.